# One Multilocus Genomic Variation Is Responsible for a Severe Charcot–Marie–Tooth Axonal Form

**DOI:** 10.3390/brainsci10120986

**Published:** 2020-12-15

**Authors:** Federica Miressi, Corinne Magdelaine, Pascal Cintas, Sylvie Bourthoumieux, Angélique Nizou, Paco Derouault, Frédéric Favreau, Franck Sturtz, Pierre-Antoine Faye, Anne-Sophie Lia

**Affiliations:** 1Maintenance Myélinique et Neuropathies Périphériques, Université de Limoges, EA 6309, F-87000 Limoges, France; corinne.magdelaine@unilim.fr (C.M.); sylvie.bourthoumieu@unilim.fr (S.B.); angelique.nizou@unilim.fr (A.N.); frederic.favreau@unilim.fr (F.F.); franck.sturtz@unilim.fr (F.S.); pierre-antoine.faye@unilim.fr (P.-A.F.); anne-sophie.lia@unilim.fr (A.-S.L.); 2Service de Biochimie et Génétique Moléculaire, Centre Hospitalier Universitaire à Limoges, F-87000 Limoges, France; 3Service de Neurologie, Centre Hospitalier Universitaire à Toulouse, F-31000 Toulouse, France; cintas.p@chu-toulouse.fr; 4Service de Cytogénétique, Centre Hospitalier Universitaire à Limoges, F-87000 Limoges, France; 5Service de Bioinformatique, Centre Hospitalier Universitaire à Limoges, F-87000 Limoges, France; paco.derouault@chu-limoges.fr

**Keywords:** multilocus disease, Charcot–Marie–Tooth, diagnosis, CNV, NGS

## Abstract

Charcot–Marie–Tooth (CMT) disease is a heterogeneous group of inherited disorders affecting the peripheral nervous system, with a prevalence of 1/2500. So far, mutations in more than 80 genes have been identified causing either demyelinating forms (CMT1) or axonal forms (CMT2). Consequentially, the genotype–phenotype correlation is not always easy to assess. Diagnosis could require multiple analysis before the correct causative mutation is detected. Moreover, it seems that approximately 5% of overall diagnoses for genetic diseases involves multiple genomic loci, although they are often underestimated or underreported. In particular, the combination of multiple variants is rarely described in CMT pathology and often neglected during the diagnostic process. Here, we present the complex genetic analysis of a family including two CMT cases with various severities. Interestingly, next generation sequencing (NGS) associated with Cov’Cop analysis, allowing structural variants (SV) detection, highlighted variations in *MORC2* (microrchidia family CW-type zinc-finger 2) and *AARS1* (alanyl-tRNA-synthetase) genes for one patient and an additional mutation in *MFN2* (Mitofusin 2) in the more affected patient.

## 1. Introduction

Charcot–Marie–Tooth (CMT) disease, the most common peripheral neuropathy, is a hereditary disorder associated to numerous genomic mutations, which can occur in different genes and in different loci of the same gene. Even though next generation sequencing (NGS) strategies, such as whole exome sequencing (WES) and whole genome sequencing (WGS), are now largely used to investigate human variations, CMT molecular diagnosis still remains difficult. Furthermore, Posey et al. showed, on a wild range of genetic pathologies, that phenotypical manifestations are the result of the combination of multiple genomic mutations in 4.9% of cases [1].

We describe here the genetic analysis of a family with two axonal CMT (CMT2) cases: Patient A (mother), characterized by axonal impairment, and patient B (daughter), with a more severe clinical condition. NGS analyses associated with Cov’Cop analysis, allowing the detection of structural variants (SV) [2], showed that both of them presented a known pathological mutation in *MORC2* (microrchidia family CW-type zinc-finger 2) and a never-described *AARS1* (alanyl-tRNA-synthetase) duplication. In addition, the more affected daughter had a third variation in *MFN2* (Mitofusin 2).

With this clinical case report, we want to highlight how CMT disease may belong to multilocus genetic pathologies. It could be relevant to take into account the possibility of a combined effect of multiple genomic mutations in order to explain the high heterogeneity of this complex clinical condition. Until now, this aspect has been poorly explored in CMT, often inducing uncompleted diagnosis and complicating the understanding of correlation between the genomic modifications and the phenotypic manifestations.

## 2. Case Presentation

This study focused on a large family with two cases of CMT2. Ethics approval was obtained from the ethic committee of Limoges University Hospital: N 386-2020-42, as well as the informed consent of all participants. This study was performed in accordance with the Declaration of Helsinki. We accessed the DNA of five members of this family who were clinically examined by a neurologist. Patients A (mother) and B (daughter) exhibited neuropathic disease phenotypes, but not individuals C, D, and E, the maternal grandmother and the maternal aunts of patient B, respectively. Patient A was a 58-year-old woman of French origin with an atypical asymmetric proximal and distal neuropathy. Her symptoms started at the age of two with gait disturbances, which progressed to a complete loss of ambulation at 43 years old. Clinically, the deficit affected the proximal and distal regions and the upper and lower limbs equally but very asymmetrically. The nerve conduction study revealed an axonal asymmetric sensory and motor neuropathy (Table 1). Median motor nerve conduction velocity (MNCV) was 40 m/s. The clinical history of patient B was slightly different from that of her mother. Patient B, a 25-year-old woman, experienced her first difficulties in walking at the age of 18 months, followed by learning problems and signs of mental deficiency in childhood. Medical examination revealed an asymmetric distal predominant sensory and motor deficit of the upper and of lower limbs, prevalent on the left side. Mild muscular atrophy was observed in both hands in association with a dystonic disorder in the finger. The examination confirmed the presence of cerebellar ataxia, with a nystagmus. There were no pyramidal signs, no diaphragmatic paralysis, no thoracic deformity, or vocal cord involvement. The asymmetry of the axonal sensory and motor neuropathy observed for patient A was also confirmed in the daughter (Table 1). Encephalic MRI showed mild vermian atrophy, without cerebellar defects. Nystagmus was also present. No clinical signs were observed in the other family members (C, D, and E).

## 3. Materials and Methods

### 3.1. DNA Extraction

Blood samples were collected in EDTA tubes. Genomic DNA was extracted by standard methods (Illustra DNA Extraction kit BACC3, GEHC).

### 3.2. Sequencing

Firstly, NGS strategy was performed on patients A and B using a 92-gene custom panel designed for CMT and associated neuropathies diagnosis (Appendix A). The amplified library was prepared with an Ion P1 HiQ Template OT2 200 kit (Ampliseq Custom, Life technologies, Waltham, MA, USA), sequenced on a Proton sequencer (Life technologies, Waltham, MA, USA), and mapped to the human reference sequence hg19/GHCh37. Secondly, for WES performed for patients A, B and E, libraries were prepared with NimbleGenSeqCapEZ-Library-SR-kits (Roche, Basel, Switzerland) and sequenced on a NextSeq-500-System (Illumina^®^, San Diego, CA, USA). Mutations of interest were verified by Sanger sequencing using forward and reverse primer pairs.

### 3.3. Bioinformatics Analysis

Variants detected by targeted NGS and by WES were annotated using the Ion reporter and Annovar software, respectively. They were evaluated with Alamut Mutation Interpretation Software (Interactive Biosoftware, Rouen, France). Databases such as ExAC Genome browser (http://exac.broadinstitute.org), dbSNP135 (National Center for Biotechnology Information (NCBI), Bethesda, Maryland, USA, (http://www.ncbi.nlm.nih.gov/projects/SNP/)), ClinVar (www.ncbi.nlm.nih.gov/clinvar) and HGMD professional (www.hgmd.cf.ac.uk) were also screened. Cov’Cop and CovCopCan, both interactive powerful software, were used to detect copy number variations (CNV) [2,3].

### 3.4. Array-Comparative Genomic Hybridization (aCGH)

Array Comparative Genomic Hybridization (aCGH) was performed using G3 Human CGH microarrays 8 × 60K (Agilent Technologies, Santa Clara, CA, USA) following the manufacturer’s instructions. Agilent CytoGenomics software (Agilent Technologies) was used to visualize, detect, and analyze copy number changes.

### 3.5. Quantitative Real-Time PCR (Q-PCR)

q-PCR reactions were carried out on genomic DNA extracted from blood samples. Primers were designed in exon 8 of the *AARS1* gene and in exon 1 of the *Albumin* gene, chosen as a reference gene. A Rotor-Gene SYBR-Green PCR Kit (400) (^©^QIAGEN, Hilden, Germany) was used following the standard protocol. Reactions were performed on the Corbett Rotor-Gene 6000 Machine (^©^QIAGEN). The Ct values of each real-time reaction were normalized, using *Albumin* as the endogenous control gene, and then compared to the normalized Ct values of three control samples. The experiment was performed in triplicates. The normalized raw data of samples were analyzed by Student’s *t*-test, comparing them with the normalized raw data of the controls. All results were statistically significant.

## 4. Results

Targeted NGS strategy revealed a heterozygous c.1403G > A mutation in the *MFN2* gene (NM_014874.3) on patient B only, resulting in the amino acidic substitution p.Arg468His. No other potentially pathological mutation was detected for patient B by targeted NGS. Sanger sequencing confirmed the presence of an *MFN2* c.1403G > A mutation in patient B, and excluded it in the other family members (Figure 1A). Given the unclear role of MFN2 p.Arg468His in CMT pathophysiology and its absence in the affected subject A, we expanded our study looking for CNV with the bioinformatics tools Cov’Cop and CovCopCan [2,3]. We detected, among the 92 sequenced genes investigated, a complete duplication of the *AARS1* gene (NM_001605.2) in both patients (ClinVar accession number: SCV001167105). *AARS1* duplication was confirmed by aCGH which allowed the identification of a 231 kb duplication, whose start and stop coordinates were identified in positions chr16:70185757 and chr16:70416579, respectively. Other genes were included in the detected duplication and they are listed in Appendix A. Although no neuropathic clinical cases caused by *AARS1* duplication have been reported, we investigated the unaffected individuals of the same family by real-time qPCR. There was no *AARS1* duplication in subject C, but it was present in unaffected subjects D and E, suggesting that *AARS1* duplication by itself is not the major cause of CMT disease of patients A and B (Figure 1B).

To elucidate the genetic cause of the disease, we performed WES on three different members of the family: subjects A, B, and E, who is the mother of six unaffected children, reinforcing the hypothesis that she does not carry a pathological mutation responsible for CMT disease in this family. Interestingly, WES data showed a heterozygous missense mutation c.568C > T (p.Arg190Trp) in the *MORC2* gene (NM_014941), which was detected in the affected individuals (A and B) but not in the healthy subject (E). c.568C > T is a known *MORC2* mutation, already described in the literature [4]. Sanger sequencing confirmed the presence of the *MORC2* mutation in subjects A and B, and its absence in subjects C, D and E. Sanger sequencing results are reported in Figure 1C. The results of the three genetic variants are summarized in Figure 2.

## 5. Discussion

The role of the three genetic variations (*AARS1* duplication, *MORC2* and *MFN2* mutations) in the clinical manifestation of CMT disease in our patients appears to be complex, but does not seem so rare according to Posey et al. who found that 4.9% of their diagnosed patients with genetic pathologies presented two or more disease loci [1].

MORC2 belongs to a family of transcriptional regulators conserved in eukaryotes and, interacting with the human silencing hub (HUSH) complex, it participates in heterochromatin regulation [5,6]. Li et al. described that, when radiation-induced double-strand breaks occur, MORC2 protein interacts with DNA repair processes to induce chromatin relaxation [7]. Concerning its pathogenic involvements, it seems that altered MORC2 expression or function could promote tumor growth, invasion, and metastasis in several cancers [8,9]. However, *MORC2* mutations are usually associated with axonal Charcot–Marie–Tooth disease type 2Z [4,10]. Classically, this *MORC2*-dependent form of CMT has an early age of onset and it is characterized by distal weakness of the lower limbs, muscular hypotonia and atrophy, foot deformities, such as pes cavus, sensory impairment, and areflexia. These clinical signs result in walking difficulties and the need of canes or a wheelchair. The *MORC2* heterozygous mutation c.568C > T (p.Arg190Trp), is, sometimes, also reported as c.754C > T (p.Arg252Trp), based on the isoform encoded by the NM_001303256 *MORC2* transcript. It was described for the first time in 2016 [4], and it appears as a hot spot, located within the GHL-ATPase domain of the MORC2 protein [4,10]. It seems to hyperactivate HUSH-mediated silencing, whereas its effect on ATPase activity remains unclear [5,11]. Moreover, in patient-derived fibroblasts, p.Arg190Trp alters the transcriptional regulation of more than 800 target genes, such as Zinc fingers (ZNFs), homeobox genes, helicases, and metallothionein genes [11]. According to our findings and previous results, the heterozygous *MORC2* mutation c.568 C > T is probably the main cause of the axonal neuropathy of patient A.

However, patient B was characterized by a more severe phenotype than patient A. This phenotypic difference may be related to the *MFN2* missense mutation c.1403G > A (p.Arg468His), which was found only in the more affected daughter (B). The *MFN2* gene encodes a mitochondrial membrane protein which plays a crucial role in mitochondrial fission and mitochondrial pathway organization. The amino acidic substitution p.Arg468His is located between the transmembrane domain and the C-terminal coiled coil region of the MFN2 protein [12]. It has already been described as the causative mutation of the axonal CMT2A, but its pathogenicity rests unclear and in the ClinVar database its interpretation is mentioned as conflicting. Engelfried et al. reported it in two patients, the first with muscular atrophy and sensory loss, the second with Parkinson’s disease and distal neuropathy, but also in an asymptomatic individual [12]. This *MFN2* variation was also found in two members of a Spanish family with a mild CMT phenotype and discrete symptoms of neuropathy [13]. Given the highly variability of the associated phenotype and its uncertain pathogenicity, functional studies were conducted on human fibroblasts carrying the p.Arg468His substitution, demonstrating a mitochondrial coupling defect and a reduced ATP production [13]. In 2011, p.Arg468His was reported to be a disease-causing mutation in association with *GDAP1* nonsense mutation p.Gln163* [14]. The clinical condition of the patient was more complex than that of her brother, who bore only the *MFN2* mutation. As well as *MFN2* and *GDAP1*, the simultaneous occurrence of two disease-causing mutations in CMT pathology (digenic inheritance) has been described for other combinations of genes, sometimes associated with intrafamilial variability [15,16,17]. This corroborates the idea that, even if the *MFN2* p.Arg468His mutation is not the primary genetic cause, it may impact the symptomatology’s severity of patient B, in our clinical case. The role of *MFN2* p.Arg468His mutation as modifier allele, in CMT, has already been suggested in a previous publication [18].

Moreover, in our study family, we detected a third variation, the complete duplication of *AARS1* gene, a SV never described before and not recorded in the GnomAD database. *AARS1* encodes the alanyl-tRNA-synthetase, the enzyme that catalyzes proper attachment of Alanine to its tRNA. In 2010, Latour et al. showed for the first time that an *AARS1* missense mutation was responsible for axonal Charcot–Marie–Tooth disease in a French family [19]. Further *AARS1* pathological mutations were then reported to be associated with CMT disease [20,21]. As well as AARS1, several other tRNA-synthases were shown to be involved in peripheral neuropathies. In these cases, dominant mutations resulted in pathological mutant proteins, and toxic gain-of-function effects, or in a protein’s loss of function [22]. However, overexpression of wildtype tRNAs has never been described to cause CMT disease, and overexpression of wildtype GARS1 in mice showed no pathological effects [23]. In our case, the presence of *AARS1* duplication in two unaffected family members, suggests that the overproduction of AARS1 enzyme does not alter the translation process and is not enough by itself to induce CMT. However, we cannot exclude that *AARS1* duplication, just like *MFN2* mutation, may modulate the phenotypic manifestation of this CMT axonal form, acting as “modifier allele”. The role of modifier alleles has been reported and analyzed in some cases of CMT disease [24,25].

## 6. Conclusions

In summary, in our study, the *MORC2* mutation (p.Arg190Trp) alone is likely responsible for axonal CMT disease (patient A). When the *MFN2* mutation (p.Arg468His) is associated with it, their effects are probably combined in a synergistic way, resulting in a more severe phenotype with additional symptoms (patient B). Lastly, an additional pathogenic role of the newly described *AARS1* duplication cannot be excluded. This genomic analysis shows how it could be complex to investigate a family clinical case if diagnosis is not complete and genetic variations are only partially detected. We believe that, for heterogeneous diseases such as Charcot–Marie–Tooth, a more accurate investigation supported by next generation sequencing technologies, would promote the discovery of new gene associations, and therefore improve the understanding of further molecular interactions and impaired mechanisms in this pathology.

## Figures and Tables

**Figure 1 brainsci-10-00986-f001:**
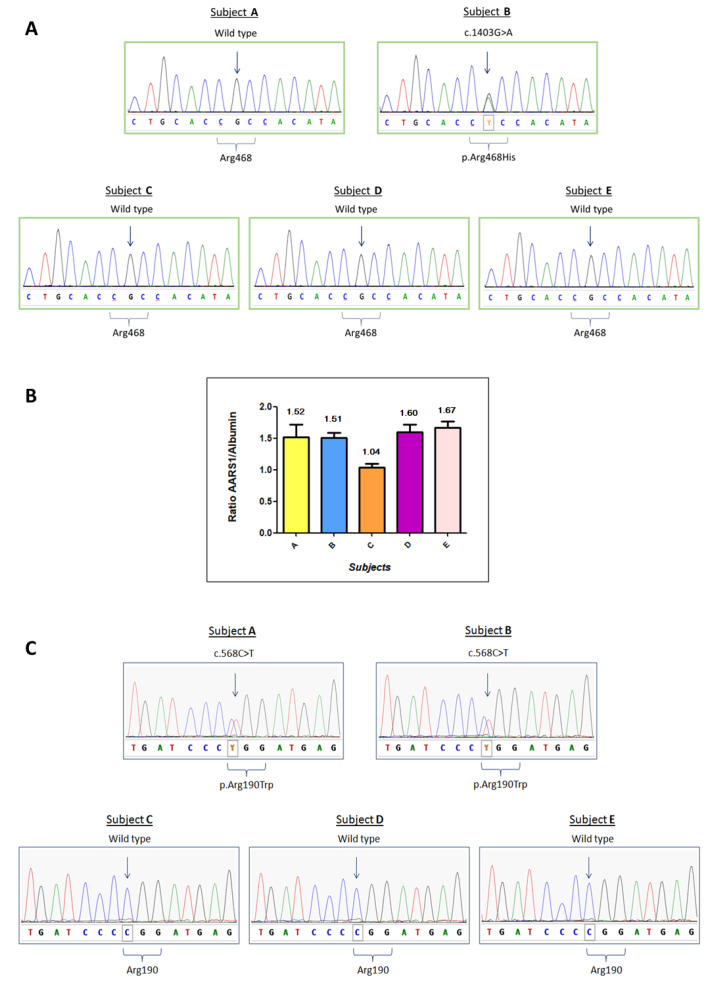
Molecular analysis of patients A to E. (**A**) Sanger Sequencing of the *MFN2* gene regarding the variation c.1403G > A, p.Arg468His. (**B**) Real-time qPCR results for *AARS1* duplication. The expected ratio is approximately 1.5 in case of duplication (three copies versus two copies) or 1 if there are no copy number variations. The plot reports the ratio means and standard deviations for each subject. (**C**) Sanger sequencing analysis of the *MORC2* gene regarding the variation c.568C > T, p.Arg190Trp.

**Figure 2 brainsci-10-00986-f002:**
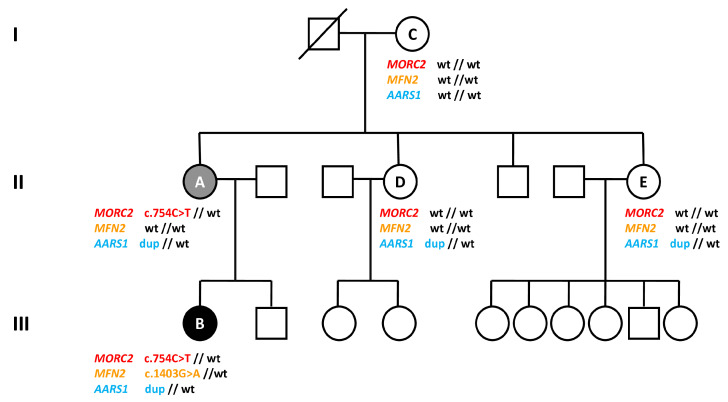
Family pedigree and summary of genetic results. The affected members are marked with black (more severe condition) or grey (less severe condition) symbols. Patients tested are indicated by letters A to E and their genotypes are specified below (wt: wildtype; dup: duplication).

**Table 1 brainsci-10-00986-t001:** Neurophysiological recordings of patient A and patient B; abnormal values are marked with bold letters or numbers (Amp: amplitude; CMAP: compound motor action potential; CV: conduction velocity; NR: no response; SNAP: sensory nerve action potential).

Subjects	Peroneal		Sural		Median						Ulnar			
	Right	Left	Right	Left	Right			Left			Right		Left	
	CMAP	CMAP	SNAP	SNAP	CMAP	CV	SNAP	CMAP	CV	SNAP	CMAP	SNAP	CMAP	SNAP
	Amp (mV)	Amp (mV)	Amp (µV)	Amp (µV)	Amp (mV)	(m/s)	Amp (µV)	Amp (mV)	(m/s)	Amp (µV)	Amp (mV)	Amp (µV)	Amp (mV)	Amp (µV)
Patient A	0.5	0.5	NR	NR	0.9	50	1.5	2.1	45	2.5	2.3	2.3	1.3	NR
Patient B	2.7	NR	NR	NR	4.7	50	3.2	3.2	45	2.8	4.1	1.2	5.7	1.0

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
