# Peer review of "One Multilocus Genomic Variation Is Responsible for a Severe Charcot–Marie–Tooth Axonal Form"

_brainsci, 2020, doi:10.3390/brainsci10120986_

Round 1
Reviewer 1 Report
The authors present an overall interesting story depicting the complexity of providing genetic diagnoses for heterogeneous diseases such as CMT. They do this by showing the results of several NGS techniques used in a single family with two affected patients, with variable severity in the phenotype.
Comments:
The authors discuss the identification of 3 different genetic variations with varying segregation in this family and provide a hypothesis about the causal role of each of these genetic variants. It would help the overall understanding of the segregation of these variants if they were depicted in the existing figure of the pedigree. Similarly it would be of help to know whether all individuals or which individuals depicted in this pedigree were assessed clinically for any signs of neuropathy.
Regarding the clinical description of the patients, it would help to know whether the proximal involvement mentioned in patient A is more dominant than distal involvement, which would be atypical for a CMT/neuropathy phenotype and whether this is different for patient B. Similarly, it would be helpful to have a more general impression of the phenotype such as axonal sensorimotor neuropathy in patient A and sensorimotor neuropathy with limited CNS dysfunction in patient B?
With regard of the AARS1 duplication it would be important to verify that the qPCR experiments were performed on genomic DNA and performed in triplicate. Also this experiment seems limited in the regard to the number of probes used both for the AARS1 gene and the control Albumin gene. It would be advised to expand these experiments and include at least 2, but preferably 3 probes for both AARS1 and Albumin. Furthermore for qPCR of gDNA one would need at least 3 control genes located on different chromosomes to compare with. Similarly this could also be used to verify the breakpoints as defined by the CGH. The mentioning of these breakpoints as detected by the CGH would also be important as it seems that while AARS1 is a CMT associated gene located within the duplication, it is not the only gene present. A (supplementary) figure depicting the genomic position of the duplication and all genes within it would be very helpful.
Regarding the identification of the MFN2 variant, it would be interesting to know whether this was the only variant that was identified in the targeted NGS screening. Also, please mention that while this variant has been reported several times as pathogenic, it is mentioned in Clinvar as conflicting evidence and that one of the earlier papers (McCorquodale et al. (2011) Mutation screening of mitofusin 2 in Charcot-Marie-Tooth disease type 2) already mentions it as rare modifying allele. Further suggestion towards its benign nature is the presence of it in gnomad with 618 heterozygous and 2 homozygous counts.
Regarding the MORC2 variant, it would be important to mention that it is a known mutation often present in literature as R190W. This also ties into the fact that in figure 3 it is mentioned as the R190W variant, but in the accompanying figure text as R252W. This variant is now also present in gnomad 5x, which is slightly high for a full pathogenic variant. While figure 3 is indeed interesting, it would be better to also include the Sanger sequencing data for the MFN2 variant to be more complete.
Textual comments:
Line 192: replace “anyway” with “however”
Reviewer 2 Report
This is an interesting case report that try to explain the possible role of multiple mutations in determining clinical variability in CMT.
Just few minor revision:
1) In the INTRODUCTION could you better describe clinical phenotypes in both A and B patients (early onset scoliosis, progressive pes cavus, diafrghmatic/respiratory impairment, vocal cord impairment, pyramidal signs such as clonus, babinsky sign, hypertonia... was any of these present in one or both patients?) Could you better explain clinical differences in motor impairment among A and B patient?
2) Could be useful in the Results to create a detailed table of neurophysiological study of all explored nerves.
3) Is it possible to correlate clinical manifestation to a gene more than other (eg asymmetrical impairment, MRI alterations)?
Round 2
Reviewer 1 Report
The authors considered the comments of this reviewer.
Now that this paper is revised, it might be relevant for the authors to inform them that Figure 1 has actually become redundant now that Figure 3 contains all the necessary information.
For the Table with NCS data it may be important that the authors indicate in bold letters which values they consider to be abnormal.
Reviewer 2 Report
Responses to comments are clear and satisfactory.
In the future it will be interesting to deep the topic about the role of genes involved in clinical manifestations.
